# Expected Labor Market Affiliation: A New Method Illustrated by Estimating the Impact of Perceived Stress on Time in Work, Sickness Absence, and Unemployment of 37,605 Danish Employees

**DOI:** 10.3390/ijerph18094980

**Published:** 2021-05-07

**Authors:** Jacob Pedersen, Svetlana Solovieva, Sannie Vester Thorsen, Malene Friis Andersen, Ute Bültmann

**Affiliations:** 1National Research Centre for the Working Environment, DK-2100 Copenhagen, Denmark; 2Finnish Institute of Occupational Health, 00032 Helsinki, Finland; 3Department of Health Sciences, Community and Occupational Medicine, University Medical Center Groningen, University of Groningen, 9713 GZ Groningen, The Netherlands

**Keywords:** longitudinal, registers, multi-state, labor market, behavioral analysis, prediction, perceived stress

## Abstract

As detailed data on labor market affiliation become more accessible, new approaches are needed to address the complex patterns of labor market affiliation. We introduce the expected labor market affiliation (ELMA) method by estimating the time-restricted impact of perceived stress on labor market affiliation in a large sample of Danish employees. Data from two national surveys were linked with a national register. A multi-state proportional hazards model was used to calculate ELMA estimates, i.e., the number of days in work, sickness absence, and unemployment during a 4-year follow-up period, stratified by gender and age. Among employees reporting frequent work-related stress, the expected number of working days decreased with age, ranging from 103 days lost among older women to 37 days lost among younger and middle-aged men. Young and middle-aged women reporting frequent work- and personal life-related stress lost 62 and 81 working days, respectively, and had more days of sickness absence (34 days and 42 days). In conclusion, we showed that perceived stress affects the labor market affiliation. The ELMA estimates provide a detailed understanding of the impact of perceived stress on labor market affiliation over time, and may inform policy and practice towards a more healthy and sustainable working life.

## 1. Introduction

The ageing workforce poses many challenges for modern societies, in terms of facilitating healthy ageing and work sustainability [1]. The increase in life expectancy, improved health of older workers, and national actions to prolong working life may lead to new, complex patterns of labor market affiliation, characterized by multiple and competing states, such as recurrent sickness absence and unemployment in between periods of work.

A multi-state approach, used in the estimation of working life expectancy, has many advantages in terms of showing the impact on lifelong labor market affiliation [2,3], when compared to the traditional Sullivan method [4]. Recent studies have used such methods to show the impact of factors like depressive symptoms, educational level, physical work demands, and occupational class on lifelong labor market affiliation [5,6,7,8].

Working life expectancy methods are based on longitudinal data, with age as the underlying time axis. In addition, these methods are used for investigating factors such as education level that will remain the same throughout the working life. No similar conceptual method has yet been developed to examine only a restricted follow-up period of labor market affiliation, using, e.g., dates or time since a major event as the underlying time axis, and for investigating factors of a temporary nature; though the idea of summing up the results of a multi-state model for restricted periods is not new [9,10,11].

This paper introduces the expected labor market affiliation (ELMA) method. The ELMA method uses a multi-state model for analyzing patterns of labor market affiliation in relation to temporary and permanent risk factors, and restricted follow-up periods, and the method additionally contains multiple possible selections for the underlying time axis. The ELMA method is useful for studying complex work participation patterns over time, and in a multi-state model containing both transitions between recurrent states (e.g., work, sickness absence, and unemployment) and transitions to absorbing states, like early retirement schemes. In the present study, we illustrate the application of the ELMA method by estimating the impact of perceived stress on labor market affiliation.

Perceived stress, defined as the degree to which situations in one’s life are appraised as stressful [12], is associated with an increased risk of sickness absence and early retirement [13,14,15,16,17]. A recent large national Danish study showed that the prevalence of perceived stress is particularly high among working adults aged 18–44 years, with a prevalence of 35% and 24% for women and men, respectively [18]. From 2010 to 2017, the prevalence of perceived stress in Denmark increased from 21% to 25%, primarily among occupationally active individuals aged 16 to 65 years [18].

Previous studies on perceived stress and labor market outcomes have investigated the association with single labor market outcomes, such as sickness absence, disability pension, or return to work [16,17]. However, in a highly flexible and dynamic labor market such as the Danish labor market, individuals are likely to experience recurrent periods of sickness absence and unemployment in between periods of work, making analysis of single outcomes an oversimplification of the complex work participation patterns [3,11,19,20].

In the present study, we aimed to estimate the impact of perceived stress on the labor market affiliation of 37,605 Danish employees applying the expected labor market affiliation method. We used a large sample of Danish employees reporting work- and personal life-related stress, linked with a register of data on labor market affiliation states. Furthermore, we investigated the possible modifying effects of gender and age on the estimation of perceived stress and on labor market affiliation.

## 2. Material and Methods

### 2.1. Study Design

The study was based on the linking of data from the Work Environment and Health in Denmark (WEHD) surveys (2012 and 2014) with longitudinal four-year follow-up data from the Danish Labor Market Accountant (LMA) register. The WEHD survey was conducted every second year from 2012 until 2018 on a large sample of Danish employees aged 18 to 64 years. In this study the WEHD 2012 wave with a response rate of 51% (*n* = 25,804) and the WEHD 2014 wave with a response rate of 57% (*n* = 29,192) were used. The total WEHD sample contained 43,209 respondents, of which 11,787 individuals responded to both waves, in 2012 and 2014.

The WEHD sample was linked to a combination of six registers from Statistics Denmark: (1) the Labor Market Accountant (LMA) register containing daily records of salary and major social benefit payments (unemployment, pensions, and sickness absence benefits, etc.) of all Danes from 2008 until 2018; (2) the education register, containing dates of the highest education level completion for all Danes; (3) the death register, with death dates for all deceased Danes; (4) the emigration and immigration register, containing dates on all emigrations and immigrations in Denmark; (5) the Register of Work Absences (RoWA), containing registrations of employment-related sickness absences down to one day of duration. The RoWA cover all public employees and a yearly weighted sample from the private sector with a coverage of approximately 37% of all private employees [17]; and (6) the employment register, containing dates on employment periods for the individuals in the RoWA register. It is important to note that in the multi-state model an individual is categorized as sick-listed, only when the sickness absence refers to the health of the individual. The RoWA register differentiates between sickness absences due to an individual’s own sickness and due to child sickness (the latter categorized as ‘temporary out’). Moreover, in the LMA register, sickness absence benefits are registered due to sickness absence of the individual only, not due to child sickness. The linked data contains individual and date-based records from 1 January 2010 to 1 December 2018 (both days included). The linkage was conducted by an encrypted version of the central person register number, given to all Danes at birth or when fulfilling the criteria for immigration during a stay of more than 3 months [21].

Of the 43,209 individuals who participated in the WEHD 2012 or/and the 2014 wave, we excluded: 377 individuals who had emigrated and did not have any registered records during the follow-up period, 4111 individuals because they did not fulfill the age criterion of 18 to 59 years at the start of the follow-up period, 903 individuals who retired prior to the start of follow-up, and an additional 213 individuals who had inconsistent or missing answers to the stress questions (*n* = 5) or reported personal life stress only (*n* = 208). The final study population consisted of 37,605 individuals (55% women). All individuals were followed for four years, starting from the individual survey answering date.

The final study population was stratified into six subsamples by gender and three age-groups: 18–39 years old (young employees), 40–49 years old (middle-aged employees), and 50–59 years old (older employees).

### 2.2. Labor Market Affiliation States

Seven states were used to model labor market affiliation during the four-year follow-up period (Figure 1). Each box in Figure 1 represents a specific, mutually exclusive labor market state and arrows represent the possible transitions. Four states are recurrent, i.e., multiple individual transitions to and from the states were possible: (1) work, periods when receiving salary; (2) sickness absence, short periods of sickness absence from one to thirty days and prolonged sickness absence periods when the employer is compensated by receiving sickness absence benefit. Sickness absence benefit compensates the employer for paying salary to sick-listed employees after the 30th day of sickness absence or is paid directly to the sick-listed when the individual is unemployed. (3) Unemployment, periods when receiving unemployment benefit as only income but available for the labor market. The unemployment benefit may rely on an insurance with a restricted duration or on the public system with no restricted duration; and (4) temporarily out of the labor market—containing all other recurrent periods of e.g., maternity leave, education, and emigration. The remaining three states are absorbing, i.e., no other transitions are possible after entering the state: (5) disability pension, time receiving a disability pension benefit, including flex-job; (6) retirement pension, time receiving an early retirement pension; and (7) death. The pension state is mostly relevant for individuals of at least 55 years at the start of the follow-up period, as the special Danish voluntary retirement scheme makes it possible to retire at 60 years. All labor market states were recorded by date.

### 2.3. Perceived Stress

Perceived stress was measured with two questions in the WEHD 2012 and 2014 surveys. First, individuals were asked “How often have you felt stressed in the last two weeks?” with answers on a Likert scale: always, often, sometimes, seldom, or never. Second, individuals were asked “What was the most important source of your stress?”, with three response categories: (1) work, (2) personal life, or (3) work and personal life. Based on the responses to both questions, participants were classified as: (1) no stress (“sometimes, seldom, or never” responses to the first question), (2) work-related stress (“always or often” responses to the first question, and “work” response to the second question), and (3) work- and personal life-related stress (“always or often” responses to the first question and “work and personal life” response to the second question). Individuals who reported personal life stress only were deleted due to the small number, and so were individuals who had inconsistent or missing answers.

### 2.4. Covariates

We included six covariates in the analysis, of which three variables were taken from the survey data: (1) working time arrangement (part-time: <37 h per week; full time: ≥37 h per week); (2) body mass index (BMI, kg/m^2^) (underweight: BMI < 18; normal weight: 18.5 ≤ BMI < 25; overweight: 25 ≤ BMI < 29.9; and obese: BMI > 29.9); and (3) smoking (yes: “daily” and “sometimes”; no: “prior smokers” and “never”). Two variables were obtained from the LMA and education registers: (4) employment sector (private/public), and (5) highest accomplished education (low/middle/high). The last variable (6) “survey year” was constructed to account for the WEHD survey: “2012”, “2014”, and “2012 + 2014”. The employment sector and highest accomplished education variables were allowed to change during the follow-up period.

### 2.5. Statistical Analyses

A separate analysis was conducted for each of the six subsamples. We followed the approach illustrated in Pedersen and Bjorner 2017 and estimated baseline instant transitions matrices for each time point from day one to day 1461, i.e., four years of follow-up with day one as the questionnaire answering date (2012 for those answering both the 2012 and 2014 wave). As the ELMA method relies on individual calculations for every combination of all covariates, the consequent grouping became too small for reliable estimations. To accommodate BMI and smoking were included in the analysis as normalized inverse probability weights, using employment sector as an equalizing variable [22].

To gain a set of instant transitions matrices corresponding to the possible combination of covariates, we adjusted the baseline matrices with a corresponding set of parameter estimates from multi-state Cox regressions. This was done for each of the six subsamples. Each Cox regression was stratified by transition and ridge regression was added to avoid overfitting [23]. In addition, the Cox regressions were adjusted for the combined weight of the inverse probability weights multiplied by the weight of the RoWA register. We used the Chapman–Kolmogorov equation (Equation (1)) for multiplying the instant transitions matrices (A(u)), to gain transition- and state-specific probabilities (P(s,t)) for each unique covariate profile in the data in the time interval from day 1 (s) to day 1461 (t). The instant state-specific probability was estimated by one minus all instant transition probabilities of leaving the state.
(1)$$\hat{P}\left(s,t\right)=\overset{t}{\underset{s}{\prod }}\left(I+\hat{A}\left(u\right)\right)$$

We estimated the transition- and state-probabilities for every unique covariate profile that we found in the six subsamples [24], including the corresponding standard deviations using the Greenwood variance for the empirical covariance matrix used in the recursion formula [25]. Next, we estimated the area under the transition probability curves and the state probabilities curves, using one hundred random resamples within the corresponding normal distribution of the mean and standard deviation. The distinct one hundred resamples of the area estimates were assigned to all individuals matching the covariate profile. The area estimation was made from the survey answering date and in days to the end of the four-year follow-up period. As the time axis, we used days from the survey answering date (day one). We treated each data record/row as a late entry and censored only at the end of the follow-up period.

The integral (E(h)), defined by the area under the probability curves, expresses the expected time spent in each state (Equation (2)). The state-specific area corresponds to the expected state duration time, given that the person was in the state from the start of the follow-up (in which: h = j, and h = (W, S, U or TO)). Likewise, the area estimate from the transition specific probability curves corresponds to the expected duration time in the “transition to state” (j), given that the person was in the initial state (h) at the start of follow-up.
(2)$$E\left(h\right)=\overset{t}{\underset{s}{\int }}P_{\mathrm{hj}}\left(s,t\right)\mathrm{du}$$

We used the Beyersmann and Putter approach [26] to gain the non-restricted duration time in each state of the model for every resample.

We used a standard analysis of variance to estimate the expected duration time for each state, with a repeated statement concerning the resamples to obtain a bootstrap estimation of the 95% confidence limits [9]. All covariates included in the Cox regression analyses were used as independent variables, and the profile and non-restricted state duration time were used as dependent variables. In addition to the gender and age-group stratified analyses, we conducted a standard analysis of variance for each subsample for each of the seven labor market affiliation states. The intercept parameter of the standard analysis of variance expresses the absolute expected state duration time in days, for an individual with variables matching the reference values. The individual parameter estimate expresses the average expected number of days. These parameters are either added or subtracted from the intercept value dependent on the value of the specific covariate [9]. We used SAS version 9.4 with the Phreg and Genmod procedure for the regression analysis, otherwise custom-made code was used to conduct all analyses.

## 3. Results

### Sample Characteristics

A total of 17,022 men and 20,583 women were included in the study. The older employees were the largest group and the middle-aged employees the smallest group (Table 1). Overall, stress was reported by 14% of the employees, ranging from 10.7% (older men) to 16.9% (young women). On average, 8% reported work-related stress and 6% work- and personal life-related stress either often or always.

Figure 1 shows the distribution of women and men at the start of the follow-up period, according to the four recurrent labor market affiliation states (work, unemployment, sickness absence, and temporarily out). The arrows illustrate the transitions during the follow-up period, with the number of transitions made by women/men and the parentheses showing the percentage of recurrent events. The majority of individuals (92% of women and 94% of men) started from a work state. Among individuals who started from sickness absence and temporarily out states there were nearly twice as many women as men. The most frequent transitions were between work and sickness absence states, followed by transitions between work and temporarily out, and between work and unemployment states. Women transitioned more frequently between work and sickness absence than men, 261,197 total transitions versus 115,165 total transitions, respectively. A total of 729 women (4%) retired from work during follow-up compared to 428 men (3%).

Table 2 and Figure 2 show the expected average number of days spent in the four recurrent labor market states (ELMA estimates) compared to employees not reporting perceived stress. The table and figure are stratified by gender and age-group.

Among individuals reporting frequent work-related stress, the expected number of working days lost increased with age, being the highest among older women (−103 days) and the smallest among younger and middle-aged men (−37 days). More lost working days were found among young and middle-aged men and women reporting frequent work and personal life-related stress than among the same aged employees reporting only work-related stress.

Among individuals reporting frequent work- and personal life-related stress, an N-shaped association between age and difference in the number of working days lost was observed, particularly for men (−46, −117, and −67 days among young, middle, and older employees, respectively). Irrespective of stress source, different shapes of associations between age and the difference in the number of sickness absence days were observed for men and women (being N-shaped and linearly increasing, respectively). Among women, the largest and smallest differences in the number of sickness absence days were seen in middle-aged employees with work-related stress (52 days) and older employees with work- and personal life-related stress (24 days), respectively. While among men, older employees with work and personal life-related stress had the largest differences in the number of sickness absence days (45 days) and young employees with work-related stress had the smallest difference (7 days).

Young and middle-aged men had more sickness absence and unemployment days when reporting frequent work and personal life-related stress than the same aged men reporting only work-related stress. In contrast, women with frequent work and personal life-related stress had fewer sickness absence and unemployment days than the same aged women reporting only work-related stress.

Young women reporting frequent work-related stress can expect 51 fewer days at work, 45 more days of sickness absence, 12 more days of unemployment, and 12 days less of being temporarily out than young women not reporting stress. Overall, the number of lost working days among individuals increased with age. While, the number of sickness absence days increased among the middle-aged and slightly declined among older women; the number of days of unemployment with the smallest difference was observed among the middle-aged women.

Among women reporting frequent work and personal life-related stress, a higher loss of working time was found than among similar aged women reporting frequent work-related stress only. This was particularly observed in young and middle-aged women.

Young men reporting frequent work-related stress can expect 37 fewer days at work, 7 more sickness absence days, and 10 more days of unemployment than similar aged men reporting rare or no stress. Middle-aged men had more sickness absence but less unemployment days, while older men had more working time lost, more sickness absence, and also more unemployment compared to the same aged men reporting rare or no stress.

Men had more sickness absence time and more unemployment time when reporting frequent work and personal life-related stress than the same aged men reporting work-related stress only. A similar comparison of middle-aged men showed that they lost more than double the working time and had more sickness absence time. For the older men the number of sickness absence days was even higher, while the working time lost was less, when compared to the same aged men reporting work-related stress only.

Appendix A show that only small differences of +/−9 days were observed in the expected time spent in three absorbing states (disability retirement, retirement pension, and death) between employees reporting and not reporting perceived stress. In addition to the perceived stress variables, Appendix A also show the ELMA estimates for the additional covariates by category.

The Appendix A are interpreted like this: the reference group consisted of individuals who reported no stress, with a middle education level, worked in the private sector in full-time employment, and participated in the 2014 WEHD survey: young women can on average expect 1227 days of work, 44 days of sickness absence, 18 days of unemployment, and 135 days of being temporarily out. Men of the same age can on average expect 1189 days of work, 29 days of sickness absence, 24 days of unemployment, 91 days of being temporarily out, and in addition to the women, on average, a death postponed by 2 days.

## 4. Discussion

In this study, we introduced the expected labor market affiliation (ELMA) method. We illustrated the use of the ELMA method by estimating the impact of perceived work-related and personal life-related stress on the average expected time spent at work, on sickness absence, and unemployment during a four-year follow-up period. We used a unique linkage of nationally representative survey data and national register data.

Overall, we found multiple changes in labor market affiliation for both genders during the four-year follow-up period and for all age groups reporting frequent stress when compared to individuals of the same gender and age group not reporting stress. Both men and women with frequent perceived stress experienced a decline in their average expected working time and an increase in time of sickness absence and unemployment. For the young and the middle-aged individuals frequent work-related stress, alone and in combination with personal life-related stress, was associated with a major decline in working time and an increase of sickness absence days. A similar, but less pronounced impact of perceived stress was seen for individuals reporting work-related stress only. In older individuals work-related perceived stress showed a greater impact than work- and personal life-related stress on the working time. While an increased time of unemployment was more frequently observed among the male employees, an increased time of sickness absence was more frequently observed among the women.

### 4.1. Advantages and Disadvantages of the ELMA Method

The ELMA method builds on multi-state methods for calculating working life expectancy or working years lost [6,7,8]. The main difference concerns the time period and underlying time axis for which estimates are calculated and the way covariates are handled. Working life expectancy estimates correspond to the expected time spent at work during a life course when having a certain age. In contrast, the ELMA estimates are calculated for a time-restricted period, which could be relatively short, and the method uses a time-axis instead of an age-axis. In working life expectancy estimations a final estimate is calculated and reported for each covariate included. The ELMA method will typically include more explainable covariates than the working life expectancy method, and an analysis of variance is used to sum up the influence of the covariates.

The ELMA method uses all information of time duration and transitions between labor market states based on the multi-state model for the entire follow-up period. Analyses with multi-state modeling of three or more states are typically characterized by a large number of transitions, and may lead to very large tables for displaying the results [11,27]. The ELMA method is able to sum up numerous results from multi-state models without compromising details on transitions, the number of states, confounders, or proportionality assumptions concerning the survival analysis part. Though the arrangement of data, the model assumptions, and the mathematical calculations in the ELMA method are comprehensive, the ELMA results are easy to interpret and communicate to stakeholders, due to their direct and absolute nature.

The ELMA method will not produce reliable estimates if there are only a few (e.g., three) individuals for any combination of covariates. It is, therefore, reasonable to use only a restricted amount of measured covariates in the ELMA method, which will depend on the sample size and the number of states. If the number of possible covariates is large, some of them could be included as inverse probability weights. In the current study, BMI and smoking were included in the analyses as normalized inverse probability weights. The length of the follow-up period, in particular, has to be considered when the multi-state model includes seldomly occurring events, e.g., transitions between work and disability pension. Plotting the transition probabilities makes it possible to inspect such “weak” events, and to assess the reliability of the curve by looking for relative smoothness. In addition, plotting the transition probabilities will make it possible to check the proportionality assumption concerning the Cox regressions. Like other working life estimations based on the multi-state model, the ELMA method is a predictive method that is based on the theoretical assumption that by cumulating the behavior of many individuals over time, an average, profile-specific behavioral pattern can be created. Such assumptions are only valid for making predictions if the underlying conditions, such as the time period and sample composition, are comparable.

### 4.2. Comparison with Previous Studies

To the best of our knowledge, this is the first study to examine expected labor market affiliation over a restricted follow-up period and including an analysis of multiple covariates. Therefore, the results of the current study cannot be directly compared with earlier studies. Typical studies related to this type of analysis use age as the underlying time axis, and an entire life-course until retirement age in terms of analyzing the working life expectation [2,3,6,7,8]. In comparison, Lie et al. (2017) used a high-dimensional multi-state model and a simple time axis in a restricted sixteen-year follow-up period from age 20 until age 55 years and showed that low IQ and mental health problems were associated with an increased risk of receiving a disability benefit [11]. Based on earlier research, showing that sex, age, socioeconomic factors, and health behaviors are associated with labor market outcomes, we included ‘educational level’, ‘smoking habits’, ‘body mass index’, and ‘working time arrangement’ [28,29,30]. Additionally, to accommodate potential selection bias we adjusted for the individual selection in terms of survey wave and adjusted for private/public sector, since the registration of sickness absence was less systematic in the private sector. The study results are in line with findings on the association between ill-health and working life expectancy. Previously, we showed a reduction in working time, an increase in time of sickness absence, and unemployment among individuals with self-reported depressive symptoms [5] and poor self-rated health [3]. Perceived stress was shown to be strongly associated with an increased risk of sickness absence for both sexes [17] and passive labor market participation (receiving sickness absence compensation, vocational rehabilitation benefits, permanent disability benefits, or unemployment benefits among 20–21 year-old men [31]). Our results suggest, not to our surprise, that frequent perceived stress at work and in personal life are particularly pronounced among female employees in the life phase of family formation. To better interpret the findings more information, e.g., on the presence of children and busy family life, is needed. Moreover, further studies should include repeated measurements of perceived stress and investigate if changes in perceived stress affect labor market affiliation.

### 4.3. Strengths and Limitations

The major strength of the present study is the large sample size with objective and detailed longitudinal register-based data on labor markets states. The use of the ELMA method allowed us to control for several confounders. However, the study limitations should also be considered when interpreting results regarding the impact of perceived stress on labor market affiliation. Data on short-term sickness absence (sickness spells less than 31 days) were not available for most employees in the private sector (approx. 63%), thus, the number of sickness absence days in the private sector was probably underestimated. Furthermore, frequent work- and personal life-related stress was assessed with two self-reported questions. Though the used perceived stress measure established an immediate condition and frequency, it might be prone to bias. Despite the large sample size, the final sample cannot be considered representative of all Danish employees. This is mainly due to the low number of young individuals in the WEHD sample [32]. Sampling weights were not used in the present study, due to considerations of not increasing the complexity of the results’ interpretation. By neglecting possible transitions between disability pension or retirement pension and death, the time on disability pension and retirement might have been overestimated. However, since the follow-up was restricted to four years and only a small fraction of individuals experienced either of the absorbing states, this overestimation may have been minimal. The results of the present study are constrained to a Danish context. The access to different labor market states depends on the rules and regulations of the labor market. Depending on the particular country, an individual with the same stress history may in one country be ‘long-term sickness absent’, but in another country with less job security be ‘unemployed’. Our study illustrates the ‘Danish case’ and if compared with other countries the differences in rules and regulations in the other countries have to be taken into account in the interpretation. The results may encourage employers to focus on work-related stress and work- and personal-related stress to decrease e.g., sickness absence. However, to find practical implications relating to stress prevention they should look elsewhere, e.g., intervention studies on stress [33].

## 5. Conclusions

As more detailed data on labor market affiliation becomes accessible, more refined methods are needed to address complex labor market affiliation patterns. This study introduced the new ELMA method to analyze complex labor market affiliation patterns, while including covariates. The application of the ELMA method for estimating the impact of perceived stress on labor market affiliation during a four-year follow-up revealed a loss in working time among Danish employees aged 18 to 59 years with frequent work- and personal life-related stress. In particular, women at the beginning of their working life or mid-career may experience a considerable loss of working time and an increased time of sickness absence if they experience frequent work- and personal life-related stress. The ELMA method contains new ways of expanding the fields of labor market, public health, and occupational health research; i.e., it can handle the complex and time varying real world information on labor market states and transitions. For instance, an individual may present with several episodes of sickness absence, before becoming unemployed and perhaps finally leaving the labor market with a disability pension. The ELMA method addresses all these labor market states and transitions in the same model, by calculating the working time loss and the corresponding time in other labor market states. The ELMA method may inform policy and practice with more detailed information about transition probabilities and labor market attachment and may help to retain individuals in work.

## Figures and Tables

**Figure 1 ijerph-18-04980-f001:**
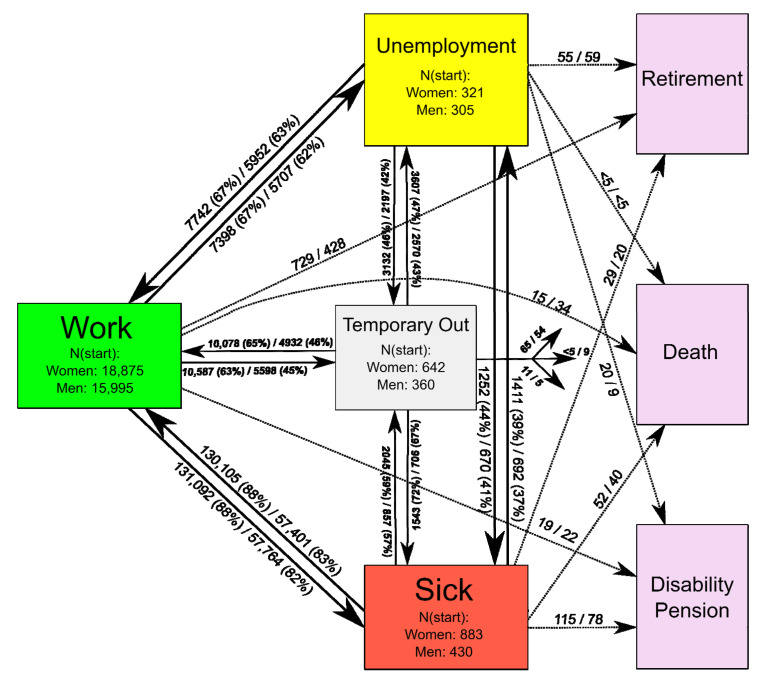
Labor market affiliation multi-state model including the descriptive illustration of the number of individuals at each state at the start of follow-up and transitions between states during follow-up. The numbers represent the number of transitions for women/men; the parentheses represent the percentages of recurrent transitions.

**Figure 2 ijerph-18-04980-f002:**
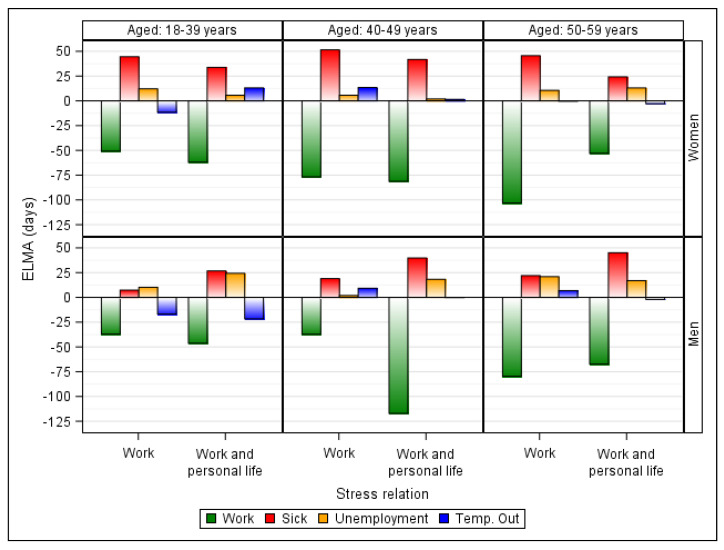
The expected average number of days spent in the four recurrent labor market states: work, sickness absence (sick), unemployment, and temporary out (temp. out). Comparison of individuals reporting perceived: work relates stress, and work and personal-life related perceived stress with individuals not reporting perceived stress. By gender and age-group.

**Table 1 ijerph-18-04980-t001:** Descriptive characteristics of the study population (*n* = 37,605).

Group	Level	Women			Men		
		Young Employees *n* (%)	Middle-Aged Employees *n* (%)	Older Employees *n* (%)	Young Employees *n* (%)	Middle-Aged Employees *n* (%)	Older Employees *n* (%)
	Total:	6782	6691	7110	5530	5428	6064
Perceived stress	No	5638 (83.1)	5649 (84.4)	5974 (84.0)	4888 (88.4)	4776 (88.0)	5416 (89.3)
	Work-related	550 (8.1)	539 (8.1)	686 (9.6)	310 (5.6)	373 (6.9)	431 (7.1)
	Work and personal life-related	594 (8.8)	503 (7.5)	450 (6.3)	332 (6.0)	279 (5.1)	217 (3.6)
Smoking	Non-smoker	4767 (70.3)	5209 (77.9)	5445 (76.6)	3934 (71.1)	4170 (76.8)	4519 (74.5)
	Smoker	1155 (17.0)	1220 (18.2)	1422 (20.0)	1225 (22.2)	1096 (20.2)	1319 (21.8)
	Not Available	860 (12.7)	262 (3.9)	243 (3.4)	371 (6.7)	162 (3.0)	226 (3.7)
BMI	Underweight	197 (2.9)	78 (1.2)	107 (1.5)	42 (0.8)	7 (0.1)	12 (0.2)
	Normal weight	3849 (56.8)	3656 (54.6)	3731 (52.5)	2755 (49.8)	2031 (37.4)	2050 (33.8)
	Overweight	1204 (17.8)	1721 (25.7)	2048 (28.8)	1788 (32.3)	2423 (44.6)	2816 (46.4)
	Obese	632 (9.3)	932 (13.9)	932 (13.1)	562 (10.2)	795 (14.6)	940 (15.5)
	Not Available	900 (13.3)	304 (4.5)	292 (4.1)	383 (6.9)	172 (3.2)	246 (4.1)
Education	Low	577 (8.5)	535 (8.0)	1207 (17.0)	821 (14.8)	704 (13.0)	1068 (17.6)
	Middle	2609 (38.5)	2884 (43.1)	2844 (40.0)	2509 (45.4)	2596 (47.8)	3002 (49.5)
	High	3572 (52.7)	3257 (48.7)	3033 (42.7)	2158 (39.0)	2089 (38.5)	1941 (32.0)
	Not Available	24 (0.4)	15 (0.2)	26 (0.4)	42 (0.8)	39 (0.7)	53 (0.9)
Employment sector	Private	2113 (31.2)	1973 (29.5)	1652 (23.2)	2806 (50.7)	2946 (54.3)	2909 (48.0)
	Public	3938 (58.1)	3805 (56.9)	4571 (64.3)	1379 (24.9)	1174 (21.6)	1695 (28.0)
	Not Available	731 (10.8)	913 (13.6)	887 (12.5)	1345 (24.3)	1308 (24.1)	1460 (24.1)
Work-time arrangement	Full-time	3711 (54.7)	4311 (64.4)	4326 (60.8)	4492 (81.2)	4939 (91.0)	5414 (89.3)
	Part-time	2175 (32.1)	2071 (31.0)	2432 (34.2)	646 (11.7)	269 (5.0)	312 (5.1)
	Not Available	896 (13.2)	309 (4.6)	352 (5.0)	392 (7.1)	220 (4.1)	338 (5.6)
Survey year	2012	2552 (37.6)	2082 (31.1)	2054 (28.9)	2025 (36.6)	1760 (32.4)	1791 (29.5)
	2014	1620 (23.9)	1902 (28.4)	2171 (30.5)	2230 (40.3)	2181 (40.2)	2411 (39.8)
	2012 + 2014	2610 (38.5)	2707 (40.5)	2885 (40.6)	1275 (23.1)	1487 (27.4)	1862 (30.7)

**Table 2 ijerph-18-04980-t002:** Expected average number of days (95% confidence interval) spent in the four recurrent labor market affiliation states (ELMA estimates) for perceived stress, stratified by gender and age group.

Gender/Age	Perceived Stress	Work	Sickness Absence	Unemployment	Temp. Out
Women					
18–39	No	. (-)	. (-)	. (-)	. (-)
	Work-related	−50.8 (−52.8:−48.8)	44.6 (43.6:45.6)	12.1 (11.7:12.6)	−11.5 (−12.2:−10.9)
	Work and personal life-related	−61.9 (−64.0:−59.9)	33.8 (32.8:34.7)	5.6 (5.4:5.9)	13.1 (12.6:13.6)
40–49	No	. (-)	. (-)	. (-)	. (-)
	Work-related	−76.7 (−79.8:−73.6)	51.5 (50.2:52.8)	5.7 (5.1:6.3)	13.5 (12.9:14.1)
	Work and personal life-related	−81.3 (−85.2:−77.3)	41.7 (40.6:42.9)	1.9 (1.5:2.4)	1.5 (0.8:2.1)
50–59	No	. (-)	. (-)	. (-)	. (-)
	Work-related	−103.1 (−105.7:−100.6)	45.6 (44.8:46.5)	10.6 (10.3:10.8)	−0.4 (−0.7:−0.2)
	Work and personal life-related	−53.0 (−55.8:−50.1)	24.2 (23.6:24.8)	13.1 (12.7:13.4)	−2.9 (−3.1:−2.7)
Men					
18–39	No	. (-)	. (-)	. (-)	. (-)
	Work-related	−37.1 (−40.4:−33.8)	7.3 (7.0:7.6)	10.2 (9.7:10.7)	−16.8 (−18.3:−15.2)
	Work and personal life-related	−46.3 (−49.4:−43.1)	26.7 (25.6:27.8)	24.3 (23.2:25.4)	−21.7 (−23.6:−19.7)
40–49	No	. (-)	. (-)	. (-)	. (-)
	Work-related	−37.2 (−39.7:−34.7)	19.0 (18.0:20.0)	1.9 (1.4:2.5)	9.2 (8.2:10.3)
	Work and personal life-related	−117.1 (−122.5:−111.7)	39.8 (38.2:41.4)	18.2 (16.8:19.7)	−0.2 (−0.8:0.3)
50–59	No	. (-)	. (-)	. (-)	. (-)
	Work-related	−79.6 (−82.1:−77.1)	22.0 (21.3:22.7)	21.0 (19.9:22.0)	6.8 (6.0:7.6)
	Work and personal life-related	−67.3 (−70.4:−64.2)	45.1 (43.3:46.8)	16.9 (15.5:18.3)	−2.1 (−2.6:−1.5)

## Data Availability

Data is available on the Researcher access at Statistics Denmark: www.dst.dk/en/TilSalg/Forskningsservice (accessed on 6 May 2021).

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
