# Peer review of "Expected Labor Market Affiliation: A New Method Illustrated by Estimating the Impact of Perceived Stress on Time in Work, Sickness Absence and Unemployment of 37,605 Danish Employees"

_ijerph, 2021, doi:10.3390/ijerph18094980_

Round 1

Reviewer 1 Report

This paper presents a very relevant approach for assessing labour market affiliation and causes for breaks and exit periods. The method seems solid and is well explained. The authors could, however, make the contribution of the approach even clearer. At least in the conclusion, they should discuss how the approach can be used for studying labour market affiliation and extending working lives in general. A few minor questions, that also require some elaboration:

Page 3, measures of perceived stress – why do you not report only personal life stress? Is this not of interest? Work and personal life-related suggests that it includes also those with “pure” work stress, but looking at the numbers in table 1 this does not seem to be the case.

Page 4 – please explain briefly why you use these covariates and not others?

Case Denmark – the one-country case study is reasonable given the extensive data you have, but you should discuss a bit more in detail the peculiarities of the Danish case. Is there any related research on other countries? Would you say your results apply also for other countries and if yes/no, how? Could institutional country-level factors explain some of your results?

Author Response

Reviewer #1:

This paper presents a very relevant approach for assessing labour market affiliation and causes for breaks and exit periods. The method seems solid and is well explained. The authors could, however, make the contribution of the approach even clearer. At least in the conclusion, they should discuss how the approach can be used for studying labour market affiliation and extending working lives in general. A few minor questions, that also require some elaboration:

Response:

We appreciate the comments and suggestions. In the conclusion section, we have added the following information to clarify the contribution of the approach:

On page 12 lines 450-453 it reads now.

“As more and more data become available, more refined methods are needed to address complex labor market affiliation patterns. This study introduced the new ELMA method to analyze complex labor market affiliation patterns while including covariates.”

And, page 12 line 462-471

“The ELMA method contains new ways of expanding the fields of labor market, public health, and occupational health research; i.e., it can handle the complex and time varying real world information on labor market states and transitions. For instance, an individual may present with several episodes of sickness absence, before becoming unemployed, and perhaps finally leave the labor market with a disability pension. The ELMA method addresses all these labor market states and transitions in the same model by calculating the working time loss and the corresponding time in other labor market states. The ELMA method may inform policy and practice with more detailed information about transition probabilities and labor market attachment and may help to retain individuals at work.”

Page 3, measures of perceived stress – why do you not report only personal life stress? Is this not of interest? Work and personal life-related suggests that it includes also those with “pure” work stress, but looking at the numbers in table 1 this does not seem to be the case.

Response:

We acknowledge that this could be clearer in the Material and Methods section. The ELMA method relies on a minimum amount of transition to occur in a sub-sample, for estimating the necessary transition probabilities. Because the number of individuals who reported solely person life related stress (possible response category 2) was small (N=208) it was not possible to perform analyses for this group (please see page 3 line 116). To clarify this, the section on “Perceived stress” page 4 line 149-156 now reads:

“(1) work, (2) personal life, or (3) work and personal life. Based on the responses to both questions, participants were classified as: (1) no stress (“sometimes, seldom or never” responses to the first question), (2) work-related stress (“always or often” responses to the first question, and “work” response to the second question), and (3) work- and personal life-related stress (“always or often” responses to the first question and “work and personal life” response to the second question). Individuals who reported personal life stress only was deleted due to a small number and so were individuals who had inconsistent missing answers.”

Page 4 – please explain briefly why you use these covariates and not others?

Response:

We have added the following information to the sub-section ‘Comparison with previous studies’ of the discussion - to explain the choice of covariates (page 11 line 398-405):

“Based on earlier research, showing that sex, age, socioeconomic factors, and health behaviors are associated with labor market outcomes, we included ‘educational level’, ‘smoking habits’, ‘body mass index’, and ‘working time arrangement’ [28-30]. Additionally, to accommodate potential selection bias we adjusted for the individual selection in terms of survey wave and adjusted for private/public sector since the registration of sickness absence was less systematic in the private sector. The study results are in line with findings on the association between ill-health and working life expectancy.”

Three new references have been added:

  1. Allebeck P, Mastekaasa A. Chapter 5. Risk factors for sick leave - general studies. Scand. J. of Public Health, 2004, 32 (63_suppl), 49-108. DOI:10.1080/14034950410021853.
  2. Tim A. Barmby, Marco G. Ercolani, John G. Treble, Sickness Absence: An International Comparison, The Economic Journal, 2002, 112, 480, F315–F331. DOI:10.1111/1468-0297.00046.
  3. Sørensen JK, Framke E, Clausen T, Garde AH, Johnsen NF, Kristiansen J, Madsen IEH, Nordentoft M, Rugulies R. Leadership Quality and Risk of Long-term Sickness Absence Among 53,157 Employees of the Danish Workforce. J Occup Environ Med, 2020, 62(8), 557-565. DOI: 10.1097/JOM.0000000000001879.

Reference number 28 should now be number 31: Trolle, N.; Lund, T.; Winding, T.N.; Labriola, M. Perceived stress among 20-21 year-olds and their future labour market participation - an eight-year follow-up study. BMC. Public Health, 2017, 17(1), 287. DOI: 10.1186/s12889-017-4179-x.

Reference number 29 should now be number 32: Arbejdsmiljodata.nfa.dk. Available online: https://arbejdsmiljodata.nfa.dk/metode.html (accessed 18 March 2021).

Case Denmark – the one-country case study is reasonable given the extensive data you have, but you should discuss a bit more in detail the peculiarities of the Danish case. Is there any related research on other countries? Would you say your results apply also for other countries and if yes/no, how? Could institutional country-level factors explain some of your results?

Response:

Thank you for raising these questions. We have added the following text to the discussion section on page 11 line 393-398:

“Typical studies related to this type of analysis uses age as the underlying time axis, and an entire life-course until retirement age in terms of analyzing the working life expectation [2,3,6-8]. In comparison to the present study, Lie et al. (2017) used a high dimensional multi-state model and a similar time axis in a restricted sixteen-year follow-up period from age 20 until age 55 years. The study by Lie et al. showed that low IQ and mental health problems were associated with an increased risk of receiving a disability benefit, but the study did not include additional covariates [11].”

Moreover, on page 12 line 439-445:

“The results of the present study are constrained to a Danish context. The access to different labor market states depends on the rules and regulations of the labor market. Depending on the particular country, an individual with the same stress history may in one country be ‘long-term sickness absent’, but in another country with less job security be ‘unemployed’. Our study illustrates the ‘Danish case’ and if compared with other countries the differences in rules and regulations in the other countries have to be taken into account in the interpretation.”

Reviewer 2 Report

The study is very interesting and refers to important and current issues of labour market. Authors used a big and reliable data set which is a strong point of the study. Population investigated  is carefully described. Please consider the following comments:

1. The study has strong methodological approach, the methods seems to be the core of the paper so I think it would be better to reflect it in the title – the audience will then expect detailed addressing the method while the conclusions and their has a minor role

2. Sickness absence – could you please explain if the data describes time of absence of an employee because of his/her sickness or absence because of the sickness of a child or any other dependant person? I think it is connected with the legal construction of the Danish system, which can be not so well known across the world. At the same time this detail can play important role for results interpretation. However, I have to admit that authors refereed somehow to this issues indicating that “To better interpret the findings more information e.g. on the presence of children and busy family life, is needed.”

3. Some practical implications for employers, social policy etc. could be more outlined. Has this model a potential to be applied for them? Or it aims purely at some scientific application.

Author Response

Reviewer #2:

The study is very interesting and refers to important and current issues of labour market. Authors used a big and reliable dataset, which is a strong point of the study. Population investigated is carefully described. Please consider the following comments:

Response:

Thank you for the constructive review, and for the comments and concerns, which were relevant and helpful.

  1. The study has strong methodological approach, the methods seems to be the core of the paper so I think it would be better to reflect it in the title – the audience will then expect detailed addressing the method while the conclusions and their has a minor role

Response:

We agree that the manuscript could benefit from a title that emphasizes the methodological approach of the study. We have therefore changed the title to (page 1 line 3-4, the change in the title is underlined):

“Expected Labor Market Affiliation: a new method illustrated by estimating the impact of perceived stress on time in work, sickness absence, and unemployment of 37 605 Danish employees”

  1. Sickness absence – could you please explain if the data describes time of absence of an employee because of his/her sickness or absence because of the sickness of a child or any other dependant person? I think it is connected with the legal construction of the Danish system, which can be not so well known across the world. At the same time this detail can play important role for results interpretation. However, I have to admit that authors refereed somehow to this issues indicating that “To better interpret the findings more information e.g. on the presence of children and busy family life, is needed.”

Response:

The raw RoWA register contains information on sickness absence, including absence due to other dependent persons. In the present study, we only included sickness absence directly related to the own health of the individual. Time of absence due to e.g., a sick child was included as a period of “Temporary Out”.  To clarify this point, we have added information to the text on page 3 line 103-109:

“It is important to note that in the multi-state model an individual is categorized as sick-listed, only when the sickness absence refers to the health of the individual. The RoWA register differentiates between sickness absences due to own sickness and due to child sickness (the latter categorized as ‘temporary out’). Also in LMA register, sickness absence benefits are registered due to sickness absence of the individual only, not due to child sickness.”

  1. Some practical implications for employers, social policy etc. could be more outlined. Has this model a potential to be applied for them? Or it aims purely at some scientific application.

Response:

We agree that the results may encourage employers to focus on work-related stress and work- and personal related stress to decrease e.g. sickness absence. However, to find solutions they will have to look at other studies, e.g. intervention studies on stress. In the present study, we aimed to apply the ELMA method using the relevant example of “stress” as an illustration to estimate labor market affiliation.

We have emphasized this point, by adding the following sentence to the limitation section of the discussion page 12 line 445-448:

“The results may encourage employers to focus on work-related stress and work- and personal-related stress in decrease e.g. sickness absence. However, to find practical implications according to stress preventions they should look elsewhere e.g. intervention studies on stress [33].“

One reference added:

  1. Egan M, Bambra C, Thomas S, Petticrew M, Whitehead M, Thomson H. The psychosocial and health effects of workplace reorganisation. 1. A systematic review of organisational-level interventions that aim to increase employee control. J Epidemiol Community Health, 2007, 61, 11, 945-54. DOI: 10.1136/jech.2006.054965.